

# Association of gender and metabolic factors with thyroid nodules in T2DM: a retrospective study

Xi Yuan[1,*], Xin Wang[1,*], Xinwen Yu[1], Yuxin Jin[1], Aili Yang[1], Xiaorui Jing[1], Shengru Liang[1], Chunni Heng[1], Na Zhang[1], Lijuan Chao[2], Langlang Liu[3], Meiying Wang[1], Yufei Liu[1], Guohong Zhao[1] and Bin Gao[1]

[1] Department of Endocrinology, Tangdu Hospital, The Fourth Military Medical University, Xi'an, China
[2] Department of Ultrasound Diagnosis, Tangdu Hospital, The Fourth Military Medical University, Xi'an, China
[3] Department of Endocrinology, Danfeng County Hospital, Shangluo, Shaanxi Province, China
* These authors contributed equally to this work.

## ABSTRACT

**Aims**. Sex differences in the incidence of thyroid nodules (TNs) are broadly recognized, but further analysis is lacking. Thus, the aim of this study was to evaluate the association between TNs and anthropometric parameters in type 2 diabetic males and females.

**Materials and Methods**. This cross-sectional study included 747 patients with type 2 diabetes mellitus (T2DM). All patients underwent clinical examination, thyroid ultrasound, laboratory tests, anthropometrics and body composition. Multivariable logistic regression assessed factors associated with TNs, and a simple nomogram was finally developed.

**Results**. In total, the incidence of TNs was 36.95% (276/747) and was significantly higher in females (52.75%) than in males (27.85%). Age was positively correlated with TNs risk in patients with T2DM (males: OR = 4.141, 95% CI [1.999–8.577], females: OR = 4.630, 95% CI [1.845–11.618]). Obesity (OR = 2.655, 95% CI [1.257–5.607]) and hyperuricemia (OR = 1.997, 95% CI [1.030–3.873]) were only associated with the risk of TNs independent of other risk factors in type 2 diabetic females, as well as other obesity factors such as weight, BMI, waist-hip ratio, percent body fat, visceral curve area, and upper arm circumference, but not in type 2 diabetic males. However, the diameter of the largest thyroid nodule was only related to age (R = 0.226, p < 0.01). Finally, the nomogram for evaluating TNs in female T2DM patients was established, and the C-index of the nomogram was 0.704 (95% CI [0.89–0.94]).

**Conclusion**. TNs occur with a significantly higher frequency in type 2 diabetic females than in males, especially those with hyperuricemia and obesity. Modifiable metabolic factors, such as obesity and hyperuricemia, are a major focus for improving TNs risk in women.

Corresponding authors
Guohong Zhao, zgh860204@163.com
Bin Gao, bingao0726@163.com

## INTRODUCTION

In recent decades, the global incidence of thyroid nodules (TNs) has risen substantially. High-resolution ultrasound, with a detection rate ranging from 20% to 76%, (*Durante et*

*al., 2018*) is currently the most sensitive method for identifying changes in nodules and detecting new formation. TNs are reported to affect 15–34.1% of Chinese adults (*Guo et al., 2014*; *Xu et al., 2021*). Although the majority of TNs discovered during routine physical examination are benign, 7–15% will progress to various types of thyroid cancer (*Wong, Farrell & Grossmann, 2018*), imposing a substantial burden on patients and society. Thus, a more complete understanding of the risk factors for TNs may help to improve prevention efforts in this population at high risk, particularly if there is evidence that modifying these risk factors can improve clinical outcomes.

Numerous studies have demonstrated a higher incidence of thyroid nodules (TNs) in elderly individuals and females (*Kobaly, Kim & Mandel, 2022*). Additionally, metabolic conditions such as obesity, diabetes, hypertension, dyslipidemia, hyperuricemia, and fatty liver have been identified as significant risk factors for TNs (*Yin et al., 2014*; *Zhang et al., 2021*; *Guo et al., 2019*; *Chen et al., 2018*). Notably, the relationship between metabolic diseases and thyroid nodules differs by gender (*Zhang et al., 2022*). For example, some studies show that metabolic syndrome and diabetes are only strongly associated with the risk of developing TNs in women but not in men (*Ding et al., 2017*), while others show that central obesity is a risk factor for TNs in men (*Zhang et al., 2021*) but not in women. However, the mechanism underlying the high incidence of TNs in women remains unknown.

Among the various metabolic risk factors, abnormalities in glucose metabolism are recognized as the most significant contributors to the development of TNs (*Gerasimova & Perova, 1985*). Reports indicate that the incidence of TNs increases progressively with worsening impairments in glucose metabolism (*Chang et al., 2021*). As one of the most common chronic metabolic diseases at present, studies have reported a 78% higher incidence of TNs and a 66% higher incidence of thyroid cancer in patients with type 2 diabetes mellitus (T2DM) compared with the normal population (*Zhang et al., 2019*). Similarly, the prevalence of diabetes in adults with TNs was found to be 24% higher than in those without TNs (*Alyousif et al., 2023*). Altogether, these studies support a "two-way relationship" between diabetes and TNs and even between thyroid cancer.

The relationship between metabolic syndrome (MetS) and thyroid morphology disorders, including thyroid nodules, can be attributed to multiple interconnected mechanisms. For example, MetS components, such as obesity, dyslipidemia, and hyperglycemia, can cause systemic chronic low-grade inflammation and oxidative stress, which, in turn, disrupt thyroid function and structure. Thyroid hormones are known to influence lipid metabolism, blood pressure regulation, and insulin sensitivity, creating a bidirectional interaction between thyroid morphology and metabolic health (*Fussey et al., 2020*). Additionally, adipose tissue accumulation, particularly visceral fat, has been linked to altered secretion of adipokines and pro-inflammatory cytokines, which may exacerbate thyroid tissue damage and nodule formation. Thyroid morphology changes, such as increased thyroid volume or nodule formation, have also been associated with MetS components like abdominal obesity and hypertension, further supporting the complex interplay between MetS and thyroid dysfunction (*Jakubiak et al., 2024a*; *Jakubiak et al., 2024b*).

Although both T2DM and female sex are recognized as risk factors for TNs, few studies have specifically focused on the risk factors for TNs in female T2DM patients, limiting the understanding of the possible mechanisms underlying the intersection between T2DM and TNs. The aim of the present study was to investigate the risk factors for TNs in male and female T2DM patients, including metabolic markers and obesity markers derived from anthropometric parameters. It is expected that elucidation of these risk factors may help facilitate earlier and more effective preventive interventions.

## MATERIALS AND METHODS

### Subjects

In this study, we retrospectively analyzed 747 adults with T2DM from hospitalized patients in the Department of Endocrinology and Metabolism of the Second Affiliated Hospital (Tangdu Hospital) of the Fourth Military Medical University, between May 2020 and March 2023 according to World Health Organization criteria (*ElSayed et al., 2023*). All included subjects underwent thyroid ultrasonography (US). Exclusion criteria included a history of thyroid surgery or medication for thyroid disease, such as thyroxine, iodine, amiodarone, antithyroid drugs, interferon (IFN), lithium, bromine salts, or immune checkpoint inhibitors (iPDL1/2). These drugs were excluded due to their significant impact on thyroid volume and structure through various mechanisms, including autoimmune thyroiditis (IFN), inhibition of thyroid hormone synthesis (lithium), and iPDL1/2 (*Mitchell, 2012*; *Ahmadieh & Salti, 2013*; *Kundra & Burman, 2012*). Other key exclusion criteria included other diabetes forms (*e.g.*, type 1 diabetes, gestational diabetes mellitus), diabetic ketoacidosis, diabetic hyperglycemic hyperosmolar state, acute cardiovascular or cerebrovascular disease (myocardial infarction or stroke), and severe hepatic dysfunction. The study was approved by the Institutional Review Board of the Medical Ethics Committee of the Tangdu Hospital,the Fourth Military Medical University of China in accordance with the tenets of the Declaration of Helsinki (No. K202207-05). All participants provided written informed consent prior to entering the study.

### Anthropometric measurements

Anthropometric measurements, including height and weight, were assessed by a well-trained physician according to a standard protocol. Body mass index (BMI) was calculated as weight/height2. The iliac waist and broadest hip circumferences were measured and used to calculate the waist-hip ratio (WHR). Upper arm circumference (AC) was measured using a tape around the thickest part of the left upper arm. Three blood pressure measurements were obtained at 30 s intervals in a seated resting state with an automated blood pressure monitor (OMRON Model HEM-752 FUZZY; Omron Company, Dalian, China), and the mean was calculated. Fasting blood samples were tested following an overnight fast and 10 min of supine rest upon arrival to evaluate biochemical parameters. Blood lipids and indices of liver and renal function, including total triglyceride (TG) and total cholesterol (TC), high-density lipoprotein cholesterol (HDL-C), low-density lipoprotein cholesterol (LDL-C), and serum uric acid (UA), were measured by using an autoanalyzer (ADVIA-1650 Chemistry System; Bayer Corporation, Leverkusen, Germany). High-performance liquid

chromatography for HbA1c was performed by using the VARIANT II Hemoglobin Testing System (Bio-Rad Laboratories, Hercules, CA, USA). TSH free triiodothyronine (fT3), and free thyroxine (fT4) levels were measured using a chemiluminescence immunoassay (Roche Cobas 8000; Roche, Rotkreuz, Switzerland).

## Measurements and ultrasound examination

Body composition was measured using digital scales with bioimpedance analysis (InBody 720; BioSpace, Seoul, Korea). Measures of body composition included percentage of body fat (PBF), waist-hip ratio (WHR), soft lean mass (SLM), fat-free mass (FFM), skeletal muscle mass (SMM), and visceral fat area (VFA). Thyroid ultrasound (R7 W, Mindray, Shenzhen, China) was performed by two experienced sonographers with subjects lying in a supine position. The size (length and width) and shape of the thyroid nodule were assessed and documented.

## Definition and diagnostic criteria

According to the 2015 American Thyroid Association (ATA) 2015 guidelines, TI-RAD class 1 is considered to indicate the absence of TNs, while TI-RAD class 2–5 is used to identify patients with TNs (*Haugen et al., 2016*). According to the study on optimal cutoff points of body mass index in Chinese adults (*Zhou, 2002*), overweight was defined as evidenced by BMI $\geq$ 24 kg/m2 and <28 kg/m2, and obese was defined by BMI $\geq$ 28 kg/m2. Abnormal systolic blood pressure was defined as SBP>130 mmHg. Hyperuricemia was defined by uric acid levels >360 $\mu$mol/L for females. Hypertriglyceridemia was defined by triglyceride levels >1.7 mmol/L.

## Statistical analysis

Significant differences between two groups (no-nodule and with-nodule groups) were calculated using either independent sample Student's *t* test (for normally distributed data) or the Mann–Whitney *U* test (for non-normally distributed data), as appropriate. The normality of continuous variables was assessed using the Shapiro–Wilk test. Univariate logistic regression analysis was performed for male and female T2DM patients, and all factors with a *P* value below 0.1 were further detected by multivariate logistic regression analysis. The correlation between maximum nodule diameter and major anthropometric parameters was determined using Spearman's correlation analysis in female T2DM patients with TNs. Finally, the predictive model for TNs was constructed, and a nomogram was depicted and used to evaluate the prediction accuracy through ROC curve analysis. and was evaluated by the concordance index (C-index) and the calibration curve. Variables that met the normality assumption were presented as means $\pm$ standard deviations, while non-normally distributed variables were presented as medians and interquartile ranges (IQRs). Absolute and relative frequencies are presented for categorical variables. Descriptive analysis was performed by SPSS 26.0, and other statistical analyses were conducted in R software (Version 4.1.2; *R Core Team, 2021*). *P* < 0.05 was considered to be a significant difference.

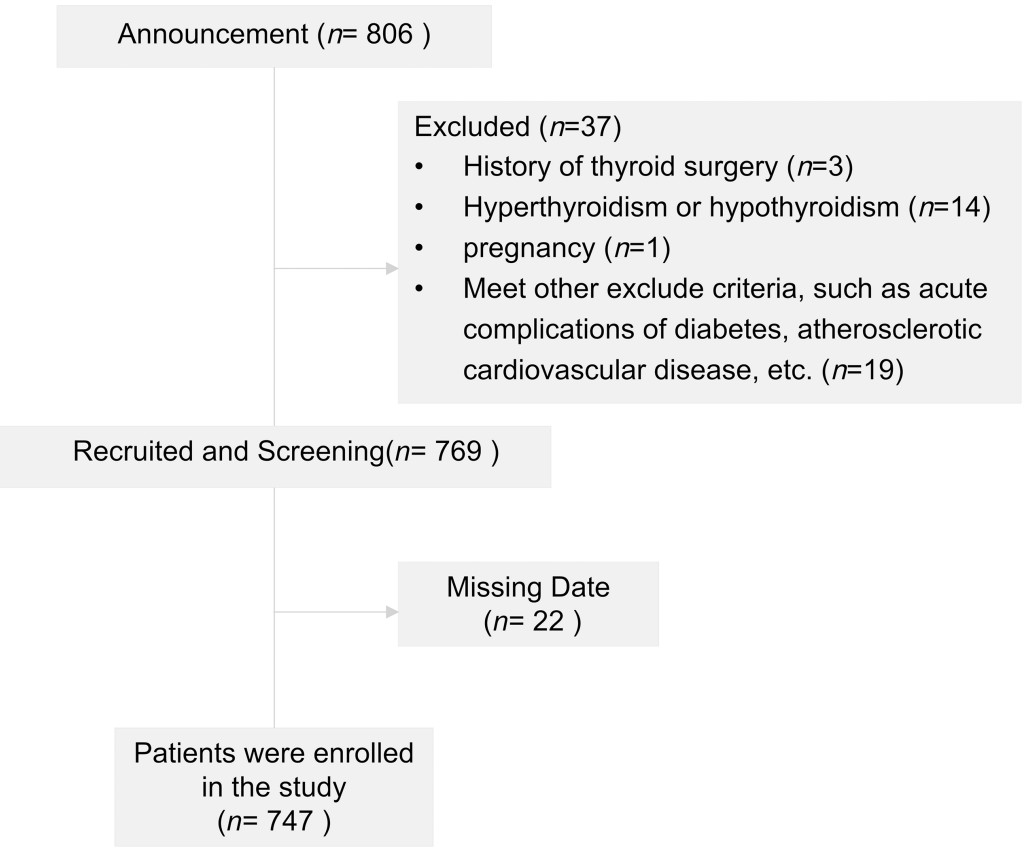

**Figure 1** Flowchart of patients included in the study.

## RESULTS

### Basic information and prevalence of thyroid nodules in the subjects

A flow diagram of the subjects is shown in Fig. 1. The baseline characteristics of male and female subjects are included in Table 1. There were 474 males (63.45%) and 273 females (36.55%), with a mean age of 54.34 ± 12.83 years. The subjects with TNs showed a high ratio of women to men (52.75% *vs.* 27.85%), and both men and women with TNs were older (males: 50.83 ± 13.02 *vs* 54.06 ± 11.4, $p < 0.001$; females: 54.06 ± 11.70 *vs* 60.78 ± 11.68, $p < 0.001$).

In male T2DM patients, the subjects in the TNs group had significantly lower HbA1c and TG levels than the subjects in the non-TNs group ($P < 0.05$), while there were no differences in blood pressure, thyroid function, body weight or body composition or other blood lipid indexes, including TC, HDL-C and LDL-C, among the groups.

SBP, UA, body weight, and other body components, including BMI, WHR, PBF, VFA, and upper AC, were significantly increased in female T2DM patients with TNs compared with non-TNs ($P < 0.05$). The incidence rate of TNs in obese females was much higher than that in patients without obesity (28.5% *vs.* 14.7%, $p = 0.006$). Overweight females also had a higher prevalence of TNs, although the difference was not statistically significant

**Table 1  Characteristics of patients with type 2 diabetes mellitus according to the presence of thyroid nodules.**

| Variables | Males | | | Females | | |
|---|---|---|---|---|---|---|
| | Unpresent $n = 342$ | Present $n = 132$ | P value | Unpresent $n = 129$ | Present $n = 144$ | P value |
| Age (years) | 50.84 ± 13.02 | 56.65 ± 11.44 | **< 0.001** | 54.06 ± 11.70 | 60.79 ± 11.68 | **< 0.001** |
| DM course (years) | 7 (2, 14) | 8 (3, 14) | 0.331 | 9 (3, 13) | 10 (3.75, 15) | 0.075 |
| Smoking (%) | 170 (49.7%) | 65 (49.2%) | 0.928 | 2 (1.6%) | 4 (2.8%) | 0.782 |
| Drinking (%) | 97 (28.4%) | 32 (24.2%) | 0.366 | 0 (0%) | 1 (0.7%) | 1.000 |
| ASCVD (%) | 101 (29.5%) | 50 (37.9%) | 0.080 | 45 (34.9%) | 65 (45.1%) | 0.085 |
| SBP (mmHg) | 131.92 ± 15.53 | 132.96 ± 15.88 | 0.514 | 129.59 ± 15.23 | 134.98 ± 17.05 | **0.007** |
| DBP (mmHg) | 84.37 ± 10.00 | 83.69 ± 8.77 | 0.471 | 79.89 ± 9.68 | 81.56 ± 10.35 | 0.171 |
| Metformin (%) | 198 (57.9%) | 85 (64.4%) | 0.196 | 69 (53.5%) | 90 (62.5%) | 0.132 |
| Insulin (%) | 153 (44.7%) | 67 (50.8%) | 0.239 | 67 (51.9%) | 65 (45.1%) | 0.262 |
| HbA1c (%) | 8.87 ± 2.24 | 8.37 ± 2.01 | **0.023** | 8.75 ± 2.11 | 8.39 ± 2.04 | 0.157 |
| UA (µmol/L) | 319.5 (270, 380) | 320 (273.75, 366.5) | 0.952 | 279 (240, 323) | 302 (243.5, 367) | **0.014** |
| TC (mmol/L) | 4.59 ± 1.29 | 4.36 ± 1.18 | 0.077 | 4.70 ± 1.21 | 4.68 ± 1.07 | 0.859 |
| TG (mmol/L) | 1.69 (1.11, 2.75) | 1.53 (1.03, 2.22) | **0.008** | 1.58 (1.05, 1.96) | 1.64 (1.19, 2.30) | 0.120 |
| HDL-C (mmol/L) | 1.04 (0.9, 1.19) | 1.02 (0.90, 1.19) | 0.888 | 1.172 ± 0.25 | 1.18 ± 0.34 | 0.877 |
| LDL-C (mmol/L) | 2.59 ± 0.95 | 2.45 ± 0.94 | 0.151 | 2.55 ± 0.93 | 2.54 ± 0.85 | 0.953 |
| TSH (uIU/ml) | 1.91 (1.17, 3.04) | 1.84 (1.27, 2.54) | 0.206 | 2.24 (1.51, 3.54) | 2.19 (1.29, 3.30) | 0.856 |
| Weight (kg) | 74.76 ± 10.89 | 76.18 ± 11.45 | 0.211 | 62.44 ± 9.29 | 65.45 ± 10.34 | **0.012** |
| Body composition | | | | | | |
| BMI (kg/m$^2$) | 25.76 ± 3.23 | 26.30 ± 3.43 | 0.107 | 24.86 ± 3.34 | 26.01 ± 3.81 | **0.008** |
| PBF (%) | 26.54 ± 6.07 | 27.38 ± 6.67 | 0.187 | 34.29 ± 5.85 | 36.23 ± 6.56 | **0.011** |
| WHR | 0.92 ± 0.06 | 0.92 ± 0.07 | 0.719 | 0.90 ± 0.05 | 0.93 ± 0.07 | **< 0.001** |
| SLM (kg) | 51.5 ± 5.95 | 51.78 ± 5.90 | 0.645 | 38.33 ± 4.44 | 38.86 ± 4.06 | 0.307 |
| FFM (kg) | 54.54 ± 6.32 | 54.84 ± 6.25 | 0.641 | 40.70 ± 4.72 | 41.27 ± 4.31 | 0.297 |
| SMM (kg) | 30.20 ± 3.78 | 30.37 ± 3.81 | 0.654 | 21.84 ± 2.79 | 22.10 ± 2.58 | 0.425 |
| VFA (cm$^3$) | 94.20 ± 33.31 | 99.12 ± 37.52 | 0.188 | 109.47 ± 37.53 | 126.33 ± 45.46 | <0.001 |
| AC (cm) | 32.32 ± 2.75 | 32.52 ± 2.85 | 0.480 | 30.19 ± 2.61 | 30.99 ± 3.07 | **0.011** |
| Overweight (%) | 232 (67.8%) | 98 (74.2%) | 0.174 | 73 (56.6%) | 96 (66.7%) | 0.087 |
| Obesity (%) | 87 | 40 (30.3%) | 0.284 | 19 (14.7%) | 41 (28.5%) | **0.006** |

**Notes.**
Abbreviations: BMI, body mass index; PBF, percentage of body fat; WHR, waist-hip ratio; SLM, soft lean mass; FFM, fat-free mass; SMM, skeletal muscle mass; VFA, visceral fat area; AC, upper arm circumference.
Bold values represent those with $P < 0.05$.

(66.7% *vs.* 56.6%, $p = 0.085$). There was no statistically significant difference between HbA1c, thyroid function, and lipid levels.

## Association between anthropometric parameters and thyroid nodule risk

In univariable logistic regression analyses, age, HbA1c, and TG were significantly associated with a high risk of TNs in males with T2DM (all $p < 0.05$; Table S1). The results of the multivariable hierarchical logistic regression analysis indicate that only advanced age (OR=4.141, 95% CI [1.999–8.577]) was an independent risk factor for TNs in males with T2DM (Fig. 2A).

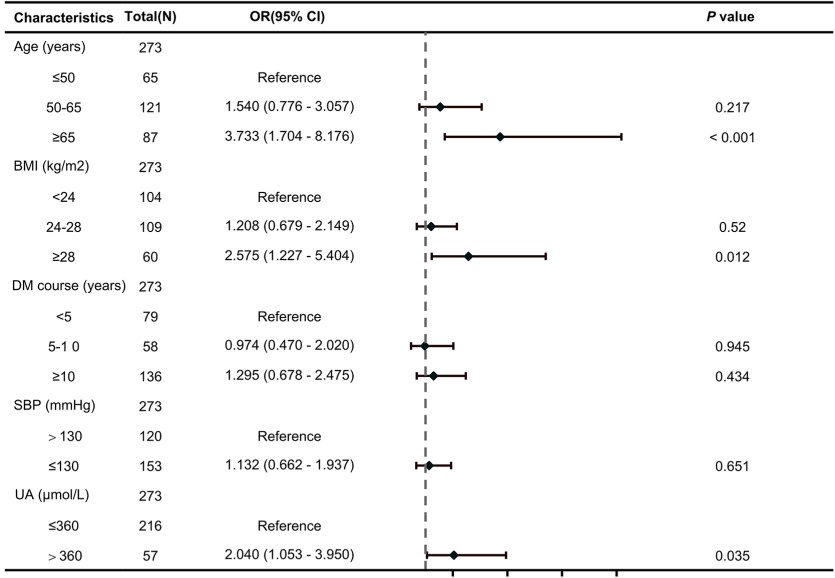

**(A)Males**

| Characteristics | Total(N) | OR(95% CI) | *P* value |
|---|---|---|---|
| Age (years) | 474 | | |
| ≤50 | 184 | Reference | |
| 50-65 | 210 | 3.249 (1.953 - 5.403) | < 0.001 |
| ≥65 | 80 | 3.305 (1.747 - 6.254) | < 0.001 |
| HbA1c (%) | 474 | | |
| <7 | 118 | Reference | |
| ≥7 | 356 | 0.764 (0.479 - 1.220) | 0.26 |
| TG (mmol/L) | 474 | | |
| <1.7 | 249 | Reference | |
| ≥1.7 | 225 | 1.010 (0.651 - 1.565) | 0.966 |

**(B)Famales**

| Characteristics | Total(N) | OR(95% CI) | *P* value |
|---|---|---|---|
| Age (years) | 273 | | |
| ≤50 | 65 | Reference | |
| 50-65 | 121 | 1.540 (0.776 - 3.057) | 0.217 |
| ≥65 | 87 | 3.733 (1.704 - 8.176) | < 0.001 |
| BMI (kg/m2) | 273 | | |
| <24 | 104 | Reference | |
| 24-28 | 109 | 1.208 (0.679 - 2.149) | 0.52 |
| ≥28 | 60 | 2.575 (1.227 - 5.404) | 0.012 |
| DM course (years) | 273 | | |
| <5 | 79 | Reference | |
| 5-1 0 | 58 | 0.974 (0.470 - 2.020) | 0.945 |
| ≥10 | 136 | 1.295 (0.678 - 2.475) | 0.434 |
| SBP (mmHg) | 273 | | |
| > 130 | 120 | Reference | |
| ≤130 | 153 | 1.132 (0.662 - 1.937) | 0.651 |
| UA (µmol/L) | 273 | | |
| ≤360 | 216 | Reference | |
| > 360 | 57 | 2.040 (1.053 - 3.950) | 0.035 |

**(C)Famales**

| Characteristics | Total(N) | OR(95% CI) | *P* value |
|---|---|---|---|
| Weight (Kg) | 273 | 1.042 (1.012 - 1.072) | 0.006 |
| BMI (kg/m²) | 273 | 1.098 (1.018 - 1.185) | 0.015 |
| WHR*100 | 273 | 1.065 (1.018 - 1.115) | 0.006 |
| PBF (%) | 273 | 1.043 (1.000 - 1.087) | 0.048 |
| VFA (cm³) | 273 | 1.009 (1.002 - 1.015) | 0.008 |
| AC (cm) | 273 | 1.135 (1.031 - 1.251) | 0.01 |

**Figure 2 Forest map of multivariate logistic regression analysis.** (A) Independent risk factors associated with TNs in male T2DM patients identified through a multivariate logistic regression analysis. (B) Independent risk factors associated with TNs in female T2DM patients identified through a multivariate logistic regression analysis. (C) Independent risk factors associated with TNs in female T2DM patients identified through a multivariate logistic regression analysis.

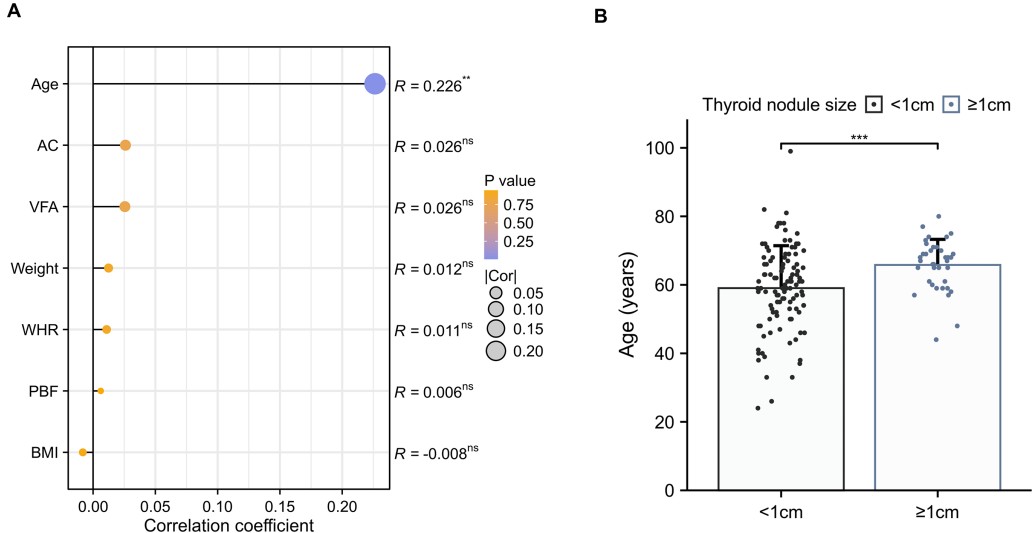

**Figure 3** **The correlation between thyroid nodule size and body composition.** (A) Spearman's correlation coefficients between the maximal diameter of the largest thyroid nodule and anthropometric parameters in females with T2DM. (B) Comparisons of age among the maximal diameter of the largest thyroid nodule (≥1 cm *vs* <1 cm) in females with T2DM.

In univariable logistic regression analyses, age, disease course, SBP, UA, obesity, BMI, WHR, PBF, VFA and upper AC were significantly associated with a high risk of TNs in females with T2DM, but skeletal muscle mass (SMM), lean body mass (SLM), or fat-free body mass (FFM) were not (Table S2). The results of the multivariable hierarchical logistic regression analysis indicate that advanced age (OR=3.733, 95% CI [1.704–8.176]), hyperuricemia (OR=1.997, 95% CI [1.030–3.873]), and obesity (OR=2.575, 95% CI [1.227–5.404]) were independent risk factors for TNs in females with T2DM (Fig. 2B). To assess the effect of obesity on TNs in more detail, multiple logistic regression showed that weight (OR=1.042, 95% CI [1.012–1.072]), BMI (OR=1.098, 95% CI [1.018–1.185]), PBF (OR=1.043, 95% CI [1.000–1.087]), WHR (OR=1.065, 95% CI [1.018–1.115]), VFA (OR=1.009, 95% CI [1.002–1.015]), and AC (OR=1.135, 95% CI [1.031–1.251]) were independent risk factors for TNs in females with T2DM (Fig. 2C).

The diameter of the largest thyroid nodule was only slightly correlated with age in females with T2DM ($R = 0.226$, $p < 0.01$) (Fig. 3A), whereas there was no association between body weight or other body composition. Among the female T2DM patients with TNs, an increase in the number of larger TNs (≥1 cm) was observed with advancing age (59.04 ± 12.38 *vs* 65.81 ± 7.46, $p < 0.001$) (Fig. 3B).

## Development and validation of an individualized prediction model

Based on multivariable logistic regression, a simple nomogram was developed for females with T2DM. The nomogram was evaluated using the receiver operating characteristic curve (ROC), calibration curve, and decision curve analysis (DCA). The final risk score was calculated by adding up the score of each item using the nomogram depicted in Fig. 4A. The term "risk" refers to the presence of TNs in females with T2DM.

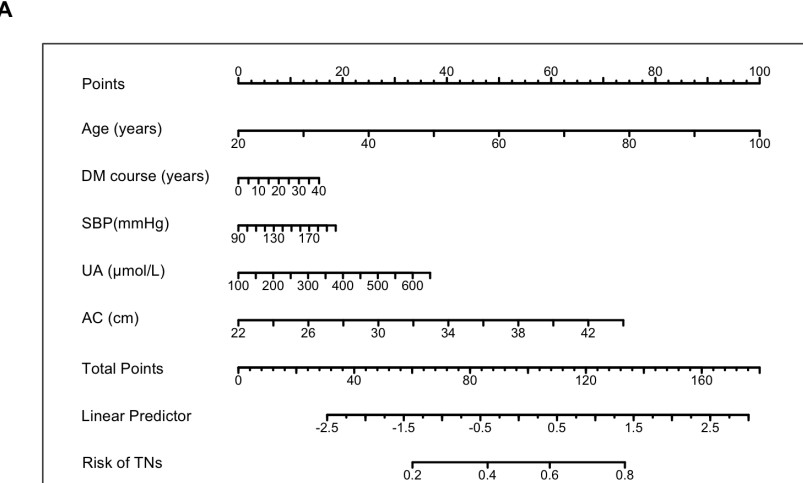

**A**

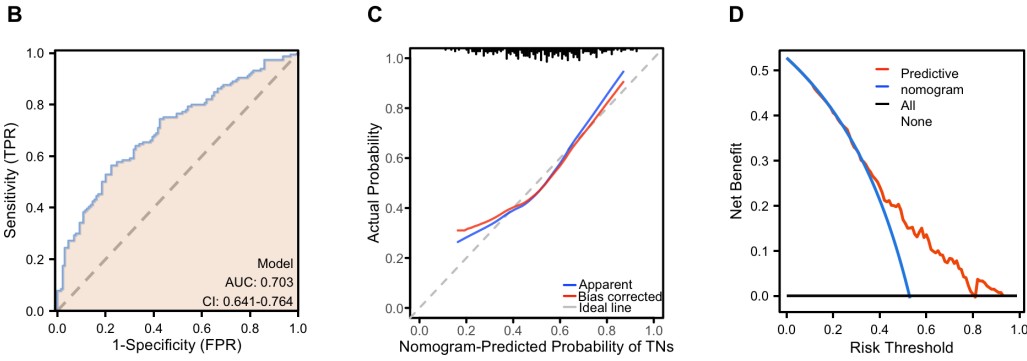

**B**     **C**     **D**

**Figure 4** **Performance and validation of the nomogram in female patients with T2DM.** (A) The nomogram calibration curve for TNs prediction model in female patients with T2DM. (B) Receiver operating characteristic (ROC) curve for the prediction model. AUC, area under the curve. (C) The calibration curve of the nomogram model. The apparent curve indicates the prediction curve, the bias-corrected curve indicates the calibration curve, and the ideal curve indicates the ideal curve. The closer the fit is to the diagonal line, the better the prediction. (D) Decision curve analysis (DCA) for the prediction model. $B = 200$ repetitions, boot. The blue line represents the predictive value, red indicates a patient predicted to have TNs, and black indicates a patient predicted to have no TNs.

ROC analysis was performed to calculate the predictive accuracy of the nomogram, and the average area (AUC) of the nomogram was 0.703 (95% CI [0.641–0.703]) (Fig. 4B). Internal validity is a way to measure if the nomogram is sound, and 200 random bootstrap resamplings were performed to reduce overfit bias. Figure 4C shows that the AUC of the random bootstrap resampling was 0.703 (95% CI [0.640–0.765]), indicating that the nomogram had satisfactory sensitivity. Furthermore, the model had good calibration by the Hosmer–Lemeshow statistic ($P = 0.4515$). The final DCA showed that if the threshold probability of patients was between 35 and 80%, screening strategies based on our nomogram' TNs risk estimates resulted in an insuperior net benefit than screen-none or screen-all strategies (Fig. 4D).

## DISCUSSION

With changes in lifestyle, the incidence of both TNs and T2DM has increased significantly worldwide in recent years. Although studies have suggested that age, gender, prediabetes, diabetes and obesity may be risk factors for TNs and even thyroid cancer (*Buscemi et al., 2018*; *Guo et al., 2014*; *Morna et al., 2022*), the risk factors for TNs in patients with pre-existing diabetes have not yet been explored. Therefore, we investigated whether obesity-related body composition indicators and related metabolic risk factors are associated with the presence of TNs in T2DM patients. Most palpable thyroid nodules are found in older women, and the incidence ratio of males to females is 1:5 (*Haugen et al., 2016*), but their clinical significance remains unclear. Based on that, differences in incidence rates by sex suggest that risk factors for TNs should be investigated separately for men and women. In the present study, we observed a TNs detection rate of 36.95% in all patients, including 52.75% for women and 27.85% for men, which was significantly higher than that in the nondiabetic population (men 23.2%, women 35.7% (*Liang et al., 2023*)). Both female sex and advanced age were identified as risk factors for TNs in T2DM patients, consistent with findings in healthy populations (*Buscemi et al., 2018*).

Obesity is significant concern as is it increases the incidence of T2DM and TNs. Previous studies have reported that 53% of adults with diabetes are obese, and the incidence of T2DM in patients with abdominal obesity was 2.14 times that in those without (*Rhee, 2020*; *Freemantle et al., 2008*). Therefore, understanding metabolic and obesity indices is critical for the developing strategies to prevent TNs in T2DM patients. Our study found that the incidence of TNs was significantly higher in obese compared to nonobese T2DM patients. Moreover, obesity remained an independent risk factor for TNs in women with T2DM after adjusting for age and other factors but not in men.

Obesity is a major contributor to insulin resistance, a common pathogenic factor for T2DM and TNs. While obesity is a significant risk factor for T2DM in both sexes, women over the age of 45 years are often overweight or tend to be more obese than men (*Beck et al., 2020*). As reported, the reduction in estrogen in older women (postmenopausal women) accelerates the development of insulin resistance (*Kim et al., 2022*). Sex differences in adiposity and insulin resistance likely contribute to the observed sex differences in the development of TNs, whereas there is no significant difference in the incidence of TNs among children who are not influenced by estrogen (*Wang et al., 2017*).

A study from China reported that crude anthropometric parameters, such as body weight and BMI, can predict the risk of thyroid nodules, especially in adult women (*Xu et al., 2015*). However, neither body weight nor BMI can differentiate between muscle and fat accumulation or adipose tissue distribution, which limits the predictability of obesity-related diseases (*Gomez-Ambrosi et al., 2012*). Recent studies have demonstrated a complex relationship between thyroid morphology, metabolic syndrome, and cardiovascular health. For example, thyroid volume and free triiodothyronine (FT3) levels have been shown to correlate with body composition parameters, such as visceral fat and waist circumference, highlighting the potential role of inflammatory and hormonal mechanisms. Moreover, subclinical cardiovascular dysfunction, assessed using markers like the ankle-brachial index

(ABI) and toe-brachial index (TBI), has been associated with thyroid morphology and function, suggesting a broader impact of thyroid abnormalities on systemic vascular health. These findings support the hypothesis that thyroid morphology and metabolic syndrome components may interact in ways that influence both cardiovascular risk and TNs, aligning with the gender-specific patterns observed in this study (*Jakubiak et al., 2024a*; *Jakubiak et al., 2024b*).

In recent years, body composition has emerged as an alternative to traditional indicators, such as PBF, VFA, and upper AC, which can assist in identifying abnormal fat distribution. Among these, upper AC has been found to be positively correlated with metabolic syndrome in a population of middle-aged and older adults (*Shi et al., 2020*) and is regarded as a simple and effective index for detecting central obesity and IR (*Zhu et al., 2020*; *Musa, Omar & Adam, 2022*). Consistent with other studies, our findings revealed of TNs in female obese patients was significantly higher than in nonobese patients. After further adjusting for age, UA and other metabolic indicators, body weight, BMI, and WHR remained independent risk factors for TNs. Furthermore, our study suggests that other obesity-related measures, including PBF, VFA, and upper AC, may serve as potential risk indicators for TNs in women with T2DM. Notably, a significant correlation between TNs and these obesity-related indicators was observed in women only, not in men. Collectively, these findings highlight the importance of identifying high-risk groups among women with T2DM and understanding the underlying sex differences, which may provide critical insights for effective management. Nevertheless, further research is warranted to elucidate the mechanisms underlying these associations.

The 2023 European Thyroid Association (ETA) Clinical Practice Guidelines for thyroid nodule management suggest that further evaluation of thyroid nodules with a diameter of ≥1 cm may be more beneficial for improving patient prognosis (*Durante et al., 2023*). Thus, more attention should be given to thyroid nodules with a diameter of ≥1 cm. Although several studies have reported that obesity is associated with thyroid nodule size, our studies showed a positive correlation between age and the maximum nodule diameter but not with obesity-related indicators. This suggests that obesity may be more associated with an increased risk of developing TNs than the risk of further development of larger nodules. We should pay more attention to TNs in elderly women with T2DM because larger nodules were consistently more prevalent in female T2DM patients at a more advanced age (*Rezzonico et al., 2011*; *Ayturk et al., 2009*). Although studies have reported that metformin use or TSH levels are associated with thyroid nodule size, our study did not find such a correlation (Table S3).

Among the relevant metabolic markers, UA is an independent risk factor for TNs in females with T2DM, even after adjustment for age, weight, and other metabolic factors. but not in men. Previous studies have shown that the correlation between UA and TNs also has gender differences, which found that uric acid has a protective effect in men over 30 years of age (*Liu et al., 2017*) but is a risk factor in both women and men under the age of 30 years. In addition, the results diverged from the findings in some previous studies, and systolic blood pressure was ultimately not identified as a risk factor independently associated with the incidence of TNs. This may be due to the limited sample size, and a

future prospective cohort study with a larger sample size should be conducted to confirm these observations.

The limitations of this study include its single-center design with a relatively small sample size, the exclusive focus on type 2 diabetes patients without including a normal population for comparison, and the lack of consideration for certain potential predictors such as dietary status and iodine nutrition. Additionally, the use of two-dimensional ultrasound limited the assessment of thyroid nodule characteristics to maximum diameter, which does not fully reflect their three-dimensional structure or biological behavior. Furthermore, due to the high proportion of missing data for anti-thyroid peroxidase (anti-TPO) and anti-thyroglobulin (anti-TG) antibody measurements, we were unable to assess their potential influence on thyroid gland size, which remains a limitation of the study. Finally, as a cross-sectional study, causal relationships between obesity and thyroid nodules cannot be established, and external validation of the nomogram's performance is still required.

## CONCLUSION

Despite well-documented sex differences, few studies have explored the variations in the clinical presentation of TNs between men and women with T2DM. Our findings indicate that obesity and hyperuricemia have a much stronger influence on TNs in women than in men with T2DM. Specifically, weight, BMI, WHR, PBF, VFA, and upper AC are independent risk factors for TNs in women with T2DM. Furthermore, nodule size is associated with age but not with obesity. These findings highlight potential metabolic mechanisms underlying sex differences in nodule formation and provide a simplified nomogram to aid in screening high-risk populations among women with T2DM. For women with T2DM, in addition to glycemic control, managing body weight is crucial for the prevention of TNs.

## ACKNOWLEDGEMENTS

The authors thank all the physicians and participants of the study for their cooperation and generous participation.

### Funding

This work was supported by the the Natural Science Foundation of Shaanxi Province, China (Grant No. 2023-JC-YB-742 and Grant No. 2022JM-438), 2020 China Diabetes Young Scientific Talent Research Project (Grant No. 2020-N-01). The funders had no role in study design, data collection and analysis, decision to publish, or preparation of the manuscript.

### Grant Disclosures

The following grant information was disclosed by the authors:
Natural Science Foundation of Shaanxi Province, China: 2023-JC-YB-742, 2022JM-438.
2020 China Diabetes Young Scientific Talent Research: 2020-N-01.

## Competing Interests

The authors declare there are no competing interests.

## Author Contributions

- Xi Yuan conceived and designed the experiments, performed the experiments, analyzed the data, prepared figures and/or tables, authored or reviewed drafts of the article, and approved the final draft.
- Xin Wang performed the experiments, analyzed the data, prepared figures and/or tables, authored or reviewed drafts of the article, and approved the final draft.
- Xinwen Yu performed the experiments, prepared figures and/or tables, and approved the final draft.
- Yuxin Jin performed the experiments, prepared figures and/or tables, and approved the final draft.
- Aili Yang performed the experiments, prepared figures and/or tables, and approved the final draft.
- Xiaorui Jing performed the experiments, prepared figures and/or tables, and approved the final draft.
- Shengru Liang performed the experiments, prepared figures and/or tables, and approved the final draft.
- Chunni Heng analyzed the data, authored or reviewed drafts of the article, data collection, and approved the final draft.
- Na Zhang analyzed the data, authored or reviewed drafts of the article, data collection, and approved the final draft.
- Lijuan Chao analyzed the data, authored or reviewed drafts of the article, data collection, and approved the final draft.
- Langlang Liu analyzed the data, authored or reviewed drafts of the article, data collection, and approved the final draft.
- Meiying Wang analyzed the data, authored or reviewed drafts of the article, data collection, and approved the final draft.
- Yufei Liu analyzed the data, authored or reviewed drafts of the article, data collection, and approved the final draft.
- Guohong Zhao conceived and designed the experiments, analyzed the data, authored or reviewed drafts of the article, and approved the final draft.
- Bin Gao conceived and designed the experiments, authored or reviewed drafts of the article, and approved the final draft.

## Human Ethics

The following information was supplied relating to ethical approvals (i.e., approving body and any reference numbers):

The Ethics Committee of Tangdu Hospital,the Fourth Military Medical University .

## Data Availability

The raw measurements are available in the Supplementary Files.

## Supplemental Information

Supplemental information for this article can be found online at http://dx.doi.org/10.7717/peerj.19068#supplemental-information.

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
