# Peer review of "Association of gender and metabolic factors with thyroid nodules in T2DM: a retrospective study"

_PeerJ, doi:10.7717/peerj.19068_

## Round 0.1 · original submission · Major Revisions

Please revise your manuscript according to the reviewers' comments.
Yours,
Yoshi
Prof. Yoshinori Marunaka, M.D., Ph.D.

Reviewer 1 ·

Basic reporting

I received for review an original research article entitled "Gender-specific associations between anthropometric parameters and Thyroid Nodules in T2DM: A cross-sectional study", prepared by Xi Yuan, which was submitted to the PeerJ. The relationship between the impact of metabolic syndrome and its components and disorders in the morphology and function of the thyroid gland, including at the subclinical level, taking into account the presence of focal changes, is a research direction that is enjoying growing interest. The research direction undertaken by the Authors is therefore interesting and valuable. It seems that research conducted in this area can contribute to a better understanding of the relationship between metabolic disorders and thyroid morphology and function. I believe that the presented manuscript presents interesting research results and has quite high scientific and cognitive value. In my opinion, however, it is worth introducing some corrections that could contribute to further improving the value and attractiveness of the manuscript.
1) I believe that the introduction should outline more precisely the pathophysiological basis for the relationship between thyroid morphology disorders, including the presence of focal lesions, and the metabolic disorders included in the criteria for diagnosing metabolic syndrome.
2) I think the description of the statistical analysis methodology should be improved. How was the conformity to the normal distribution tested?
3) I believe that it is worth citing a series of two recently published articles in the discussion, in which the same population demonstrated certain relationships between thyroid morphology and function and body composition and the presence of metabolic syndrome, as well as subclinical cardiovascular dysfunction assessed using the ankle-brachial index and the finger-brachial index. These studies show that thyroid morphology and function disorders in the course of metabolic syndrome and its components may also be related to the pathogenesis of cardiovascular diseases. (10.3390/medicina60091445; 10.3390/medicina60071080).

4) The text should be revised for minor editorial, linguistic and stylistic corrections.

Experimental design

No additional comments.

Validity of the findings

No additional comments.

Additional comments

No additional comments.

Reviewer 3 ·

Basic reporting

I appreciate the opportunity to review  the manuscript entitled "Gender-specific associations between anthropometric parameters and Thyroid Nodules in T2DM: A cross-sectional study".

The article uses understandable and professional language that does not raise any objections. References to previous scientific works covering the issue discussed by the authors are correct, and no unnecessary self-citations were noticed.

The article has a structure characteristic of original works. The raw data are accessible, and I evaluate their utilization as accurate. The work does not provide issues pertaining to suspected plagiarism.

Experimental design

In the methodological section, my concerns pertain to the following issues:
1. In lines 88-89, the authors stated that the exclusion criterion was the use of drugs affecting thyroid function - it is essential to provide a rationale for why certain listed substances are contraindicated while others are not, noting that substances such as IFN, lithium, and bromine salts, or iPDL1/2, can also influence the volume and structure of the thyroid gland. 
2. In lines 107-108, it is necessary to elaborate on the abbreviations HDL-C and LDL-C. 
3. The authors assessed TSH concentration (lines 111-112); however, the omission of fT4 and/or fT3 assessments raises questions regarding the completeness of the clinical picture, as well as the exclusion of anti-TPO and anti-TG antibody levels, which may significantly impact thyroid gland size, particularly in the presence of concomitant diabetes.
4. In lines 118-119, the authors indicate that two dimensions of nodules in the thyroid gland were evaluated: length and width - given that changes are three-dimensional, the evaluation must also consider height / depth. Please provide your insights on this observation.
5. In the section on statistical analysis, the authors describe the use of parametric and non-parametric tests quite freely (for example, in lines 129-130 the Student's t-test and U-Mann Whitney) - please expand the information on the tests used to assess the distribution of variables. In the case of using non-parametric tests, it also seems more justified to use the median and interquartile range for the assessment of variables than the mean and standard deviation.

Validity of the findings

The results section of the study does not present significant issues; nonetheless, the observations regarding the methodology, particularly those related to the statistical analysis and ultrasound evaluation of alterations in the thyroid gland, warrant consideration.

The discussion is conducted in a correct, interesting way, arousing the reader's interest. As I mentioned earlier, I have no comments on the references. The conclusions are clear from the results and are correct, provided that the aspects concerning potential methodological errors described above are well explained.

I congratulate the authors of the work on the effort they have undertaken - I believe that the article has great potential, but requires major revision, especially in the methodological area. The errors and understatements contained therein raise justified doubts about the need to modify the rest of the article.

---

## Round 0.2 · Minor Revisions

Please revise your manuscript according to the reviewer's comments.

Yours,
Yoshi
Prof. Yoshinori Marunaka, M,D, Ph.D.

Reviewer 1 ·

Basic reporting

I received for review a revised version of the original research article entitled "Gender-specific associations between anthropometric parameters and Thyroid Nodules in T2DM: A cross-sectional study", prepared by Xi Yuan et al., which was submitted to the PeerJ. In my opinion, the paper has been significantly improved. I have no further critical comments. I recommend the manuscript for publication in its current form.

Experimental design

No additional comments.

Validity of the findings

No additional comments.

Additional comments

No additional comments.

Reviewer 3 ·

Basic reporting

I appreciate the opportunity to second review the manuscript entitled "Gender-specific associations between anthropometric parameters and Thyroid Nodules in T2DM: A cross-sectional study" after revision.

As I mentioned before, I appreciate the opportunity to review the manuscript entitled "Gender-specific associations between anthropometric parameters and Thyroid Nodules in T2DM: A cross-sectional study".

References to previous scientific works covering the issue discussed by the authors are correct, and no unnecessary self-citations were noticed.

The article has a structure characteristic of original works. The raw data are accessible, and I evaluate their utilization as accurate.

The work does not provide issues pertaining to suspected plagiarism.

Experimental design

uthors appropiately answers to my comments about the need to list specific substances that constitute a contraindication to including patients' data in the study. Similarly, they appropriately addressed the comment regarding elaborate on the abbreviations of HDL-C and LDL-C. I also have no comments on the answers regarding statistical analysis.

In response to my comments (included in a separate file), the authors responded to the accusation regarding the omission of fT4 and fT3 measurements - it was pointed out that due to the lack of statistically significant differences, these data were not included in the main manuscript. Personally, I believe that these data should be found at least in the supplementary files, for readability and transparency.

Also in response to my comments contained in a separate file, the authors replied that they indicated as a limitation of the study the information about the lack of antibody markings - and yet, ultimately, I do not find such content of the statement in revised manuscript.

In response to the comment regarding the missing dimensions of thyroid nodules, the authors indicated this as a limitation of the study, but at the same time included in their response the information that the primary focus of their study was to assess the impact of gender differences and metabolic risk factors on the occurrence, not size of thyroid nodule. In such a situation, it would seem justified to clarify this by changing the title of the manuscript.

Validity of the findings

As I mentioned before, the discussion is conducted in a correct, interesting way, arousing the reader's interest. As I mentioned earlier, I have no comments on the references. The conclusions are clear from the results and are correct, provided that the aspects concerning potential methodological errors described above are well explained.

Additional comments

I congratulate the authors of the work on the effort they have undertaken - I believe that the article has great potential, but still requires some revisions.

---

## Round 0.3 · accepted · Accept

Congratulations!

Thank you for your submission.
Yours,
Yoshi
Prof. Yoshinori Marunaka, M.D., Ph.D.

Reviewer 3 ·

Basic reporting

I have no further critical comments. I recommend the manuscript for publication in its current form.

Experimental design

I have no further comments.

Validity of the findings

I have no further comments.

Additional comments

I have no further comments.